# A Buried Water Network Modulates the Activity of the *Escherichia coli* Disulphide Catalyst DsbA

**DOI:** 10.3390/antiox12020380

**Published:** 2023-02-04

**Authors:** Geqing Wang, Jilong Qin, Anthony D. Verderosa, Lilian Hor, Carlos Santos-Martin, Jason J. Paxman, Jennifer L. Martin, Makrina Totsika, Begoña Heras

**Affiliations:** 1Department of Biochemistry and Chemistry, La Trobe Institute for Molecular Science, La Trobe University, Melbourne, VIC 3086, Australia; 2Centre for Immunology and Infection Control, School of Biomedical Sciences, Queensland University of Technology, Brisbane, QLD 4000, Australia; 3Griffith Institute for Drug Discovery, Griffith University, Nathan, QLD 4111, Australia

**Keywords:** thioredoxin, redox regulation, disulphide, protein folding, bacterial pathogenesis

## Abstract

The formation of disulphide bonds is an essential step in the folding of many proteins that enter the secretory pathway; therefore, it is not surprising that eukaryotic and prokaryotic organisms have dedicated enzymatic systems to catalyse this process. In bacteria, one such enzyme is disulphide bond-forming protein A (DsbA), a thioredoxin-like thiol oxidase that catalyses the oxidative folding of proteins required for virulence and fitness. A large body of work on DsbA proteins, particularly *Escherichia coli* DsbA (EcDsbA), has demonstrated the key role that the Cys^30^-XX-Cys^33^ catalytic motif and its unique redox properties play in the thiol oxidase activity of this enzyme. Using mutational and functional analyses, here we identify that a set of charged residues, which form an acidic groove on the non-catalytic face of the enzyme, further modulate the activity of EcDsbA. Our high-resolution structures indicate that these residues form a water-mediated proton wire that can transfer protons from the bulk solvent to the active site. Our results support the view that proton shuffling may facilitate the stabilisation of the buried Cys^33^ thiolate formed during the redox reaction and promote the correct direction of the EcDsbA–substrate thiol–disulphide exchange. Comparison with other proteins of the same class and proteins of the thioredoxin-superfamily in general suggest that a proton relay system appears to be a conserved catalytic feature among this widespread superfamily of proteins. Furthermore, this study also indicates that the acidic groove of DsbA could be a promising allosteric site to develop novel DsbA inhibitors as antibacterial therapeutics.

## 1. Introduction

Cysteine is one of the least abundant amino acids in proteins. The negative selection of cysteines stems from the high reactivity of free thiols (R-SH), which in oxidising environments such as the extracellular space, the bacterial cell envelope or the eukaryotic endoplasmic reticulum, is highly susceptible to oxidative damage, such as the formation of sulfenic acid or non-native disulphide bonds [1]. Consequently, in bacteria, cysteines are often found in even numbers in exported proteins to facilitate the formation of disulphide bonds and leave no unpaired cysteines [2]. By providing stability and increasing resistance to proteases, disulphide bonds in proteins are critical for the structural integrity and function of many secretory proteins. Although a pair of cysteines can slowly form a disulphide bond by air oxidation, most prokaryotes and eukaryotes encode a disulphide bond (DSB) machinery, consisting of enzymes belonging to the thioredoxin (TRX) superfamily that catalyse disulphide bond formation, reduction and isomerisation [3,4,5].

The thiol oxidase DsbA is the master disulphide catalyst in a wide range of bacteria [6,7,8,9,10,11,12,13,14]. This protein is required for folding bacterial proteins involved in key virulence mechanisms such as motility, host cell manipulation and colonisation [15]. Importantly, bacterial pathogens with a dysfunctional DSB system exhibit reduced virulence in animal infection models [16,17,18,19]. DsbA from *E. coli* K−12 (EcDsbA) was the first and is probably the best-characterised DSB enzyme to date [3]. This 21 kDa periplasmic protein is a dithiol oxidase that catalyses disulphide bond formation between consecutive cysteines of unfolded polypeptides as they are translocated to the periplasm [20]. EcDsbA consists of a canonical TRX domain [21], a hallmark of proteins involved in redox homeostasis, with an inserted helical domain [21] (Figure 1A). The TRX domain harbours the conserved active site CXXC (C30-P31-H32-C33 in EcDsbA) that is featured in many proteins of the TRX superfamily [22]. The strong dithiol oxidase properties of EcDsbA are primarily attributed to the nucleophilicity of C30 [23]. In the disulphide bond exchange reaction, the solvent-exposed *N*-terminal C30 of oxidised DsbA reacts with reduced substrates and directly forms a mixed-disulphide intermediate [15]. This covalent intermediate is resolved to release the oxidised substrate and reduced DsbA. The latter is then recycled by the inner membrane protein DsbB in concert with its cofactor ubiquinone, restoring DsbA to its active oxidised state for the next redox cycle [24].

The EcDsbA crystal structure [21] revealed several prominent surface features, including a hydrophobic groove and a hydrophobic patch neighbouring the catalytic site (Figure 1B). Further studies demonstrated that those regions corresponded to the binding sites for the redox partner DsbB [25] and substrate proteins [26], respectively. The structure of EcDsbA also shows an acidic groove located in the opposite non-catalytic face of the protein [21] (Figure 1C). The acidity of this groove originates from a cluster of negatively charged residues (E24, E37, E38, D44, E85, D86 and E94) that map along a buried channel in the interface between the TRX and helical domains of EcDsbA (Figure 1C). Since the pioneering studies on EcDsbA, other DsbA proteins from various bacteria have been structurally investigated [7,9,10,27,28,29,30,31]. These studies showed that although all proteins share the same overall architecture, the long hydrophobic groove and the hydrophobic patch adjacent to the CXXC motif are not strictly conserved among DsbA enzymes [6]. Conversely, the acidic region at the back of the EcDsbA catalytic site appears to be a commonly shared feature. Despite the high conservation of the acidic groove in DsbA proteins, its function remains mostly unknown.

In this work, we employed site-directed mutagenesis combined with biochemical and structural analyses and cell-based assays to investigate the charged residues lining the buried channel of EcDsbA. Our work shows that replacement of these charged residues with alanine does not affect the redox properties of EcDsbA that typically define its oxidase activity. Conversely, through functional assays, we demonstrate that the introduced mutations alter the catalytic activity of this thiol oxidase. Our structural data suggest that the conserved charged/polar residues neighbouring the buried catalytic cysteine may be required for stabilising a water-mediated proton wire that transfers protons from the bulk solvent to the active site cysteines to facilitate DsbA-mediated disulphide bond catalysis.

## 2. Materials and Methods

### 2.1. Generation of DsbA Mutants

Wildtype *E. coli* DsbA with the predicted signal peptide sequence was cloned into a pBR322 vector to generate a periplasmic expression vector. DsbA mutants were generated using the Stratagene QuikChange II method with Q5 High-Fidelity DNA polymerase (NEB) and DpnI (NEB). Oligonucleotides were obtained from Sigma Aldrich. The following oligonucleotides were used for site-directed mutagenesis: E24A, 5’-GCA AGT GCT GGC GTT CTT CTC TTT CTT−3’; E37A, 5’-CTG CTA TCA GTT TGC CGA AGT TCT GCA TAT T−3’; K58A, 5’-GTG AAG ATG ACT GCA TAC CAC GTC AAC T−3’. Successful mutations were verified by dideoxynucleotide sequencing (Macrogen, Republic of Korea). The wildtype and mutant DsbA proteins were expressed in *E. coli* BL21(DE3), grown in autoinduction media. After harvesting the cells, periplasmic proteins were extracted by resuspending the cell pellet with lysis buffer (20 mM Tris-HCl, pH 8.0, 25 mM NaCl, 2–4 mg/mL colistin sulphate) and incubating for 18–24 h with gentle stirring at 4 °C. After centrifugation, EcDsbA was purified from the supernatant using a three-step purification method, including hydrophobic interaction chromatography, anion exchange chromatography and size-exclusion chromatography. DsbA was oxidised with 10 molar excess copper (II) phenanthroline (Cu(II)Phen_3_) and buffer exchanged to the final storage buffer (20 mM HEPES, pH 7.0).

### 2.2. Determination of Redox Potential

Oxidised EcDsbA mutants (3 µM) in a degassed buffer of 10 mM of sodium phosphate, pH 7.0, 0.1 mM EDTA, were incubated with 1 mM of oxidised glutathione (GSSG) and a range of reduced glutathione (GSH) concentrations (0.01 µM−1 mM) for 16–20 h at 25 °C. The fluorescence of DsbA mutants was measured in a 96-well half-area black plate (Geiner) with a fluorescence excitation wavelength at 280 nm and an emission wavelength at 332 nm. The fluorescence of protein was plotted against the ratio of [GSH]^2^/[GSSG], and the equilibrium constant, K_eq_, was calculated using a binding equation: Z = ([GSH]^2^/[GSSH])/(K_eq_ ([GSH]^2^/[GSSH])), where Z is the fluorescence of the sample at equilibrium. The equilibrium constant, K_eq_, and redox potential can be determined from the Nernst equation: E^0’^= E^0’^_GSH/GSSG–_(RT/2F) × ln K_eq_, where E^0’^_GSH/GSSG_ is the standard potential of −240 mV, R is the universal gas constant 8.314 J K^−1^ mol^−1^, T is the absolute temperature in K, F is the Faraday constant 9.648 × 10^4^ C mol^−1^, and K_eq_ is the equilibrium constant. *K*_eq_ is calculated as mean ± standard error of the mean (SEM) with at least three independent replicates.

### 2.3. Determination of pK_a_ Values

The p*K*_a_ values of the nucleophilic C30 in EcDsbA mutants were determined by monitoring the specific absorbance of the thiolate anion at 240 nm. Measurements were carried out for both oxidised and reduced EcDsbA at room temperature at 32 different pH values, ranging from 2.5 to 7.5, in a buffer system consisting of 10 mM of dipotassium phosphate, 10 mM of monopotassium phosphate, 10 mM of sodium citrate, 10 mM of Tris, 1 mM of EDTA and 200 mM of KCl. Oxidised and reduced EcDsbA were prepared by incubating with 10 molar excess Cu(II)Phen_3_ or DTT, respectively. After removing the oxidising and reducing reagents, the redox state of proteins was confirmed by Ellman’s reagent. The absorbance of samples at 240 nm and 280 nm was measured in a 96-well UV star microplate (Greiner) using the Fluorostar Omega microplate reader (BMG Labtech). The pH dependence of the thiolate-specific absorbance signal (S(A_240_/A_280_)_reduced_/(A_240_/A_280_)_oxidised_) was fitted according to the Henderson–Hasselbalch equation. p*K*_a_ is expressed as mean ± standard error of the mean (SEM) with at least three independent replicates.

### 2.4. Circular Dichroism Spectroscopy

CD spectroscopy was performed using an AVIV 410-SF CD spectrometer. Wavelength spectra were collected between 190 and 250 nm for 0.2 mg/mL DsbA mutants in a buffer of 100 mM of sodium phosphate, pH 7.0, 50 mM of NaCl and 1 mM of EDTA using 1 mm quartz cuvettes with a step size of 1 nm. The same sample was also used to determine the melting temperature (T_m_) of DsbA mutants by collecting CD spectra at 222 nm as a function of temperature (40–90 °C). Data were analysed and plotted using GraphPad Prism 8.

### 2.5. Peptide Oxidation Assay

The thiol oxidation activity of DsbA mutants was determined using a fluorescently labelled peptide derived from PilQ [10] or ASST (Purar Chemicals, Australia). In brief, the peptide substrate (CQQGFDGTQNSCK or CNENGLCK with a europium DOTA (1,4,7,10-tetraazacyclododecane−1,4,7,10-tetraacetic acid) at the N-terminus and a methylcoumarin amide group coupled to the lysine at the C-terminus) was prepared at 2 mM in 100 mM of imidazole, pH 6.0. A standard 50 µL reaction mixture consists of 0.5 µM of DsbA, 2 mM of oxidised glutathione and 16 µM of substrate peptide in a buffer of 50 mM of MES, pH 5.5, 50 mM of NaCl and 2 mM of EDTA. Assays were performed in a PerkinElmer 384-well white opaque microplate (OptiPlate−384). The time-resolved fluorescence of substrate peptide oxidation (excitation λ = 340 nm and emission λ = 615 nm) was measured using a CLARIOstar plate reader (BMG Labtech) fitted with the TR-FRET module. Measurements were carried out in triplicate for each protein. Data were analysed and mean ± standard error of the mean (SEM) was plotted using GraphPad Prism 8 (GraphPad Software, Inc., San Diego, CA, USA).

### 2.6. Kinetics of Oxidation of PapD Peptide by EcDsbA Mutants

The kinetics of oxidation of the PapD peptide Ac-FICNFSRCSV-CONH_2_ (Mimotopes, Australia) by DsbA mutants was followed by monitoring the change in intrinsic fluorescence of EcDsbA upon reduction at wavelengths > 320 nm (excitation λ = 295 nm) using an Applied Photophysics SX20 stopped flow apparatus, thermostatted at 25 °C. Since EcDsbA contains a tryptophan residue spatially close to the active site cysteines, the intrinsic fluorescence of the protein is influenced by its redox state. The peptide derived from PapD does not contain any tryptophan residues and is therefore fluorescently silent. This situation allowed monitoring of the oxidation of PapD by DsbA mutants via the increase in tryptophan fluorescence change of DsbA upon reduction. Mutations slightly changed the intrinsic fluorescence of EcDsbA, and this has been corrected for in the normalisation of the data (Appendix A). Both peptide and protein samples were prepared in a degassed and sterile filtered buffer of 20 mM of HEPES, pH 7.0, 100 mM of NaCl and 0.1 mM of EDTA. Measurements were carried out using a 1:1 volume mixing ratio under second-order reaction conditions with initial concentrations of both DsbA and PapD peptide at 1 µM. The initial rates of the oxidation reactions were determined from the linear portion of the fluorescence data using Pro-Data SX (Applied photophysics), and these were used to determine the apparent second-order rate constant for the reaction. Initial rates and rate constants are calculated as mean ± standard error of the mean (SEM) with at least three independent replicates.

### 2.7. Electron Transfer Experiments

Oxidised DsbA was prepared by incubating protein with 10 molar excess Cu(II)Phen_3_ at 25 °C for 1 h and the sample was then buffer-exchanged into 20 mM of HEPES, pH 7.0, 100 mM of NaCl and 0.1 mM of EDTA. The oxidised state of DsbA was confirmed by the Ellman assay. For the electron transfer assay, 25 µL of 20 µM of DsbA was mixed with 25 µL of 22 µM of PapD-derived peptide at a molar ratio of 1:1.1. The reaction was quenched at 10 s, 20 s, 30 s and 120 s with 10% (*w*/*v*) TCA. After centrifugation, the pellets were washed with cold acetone and solubilised in a buffer of 50 mM of Tris-HCl, pH 7.0, 1% SDS containing 4 mM of AMS (Life Technologies, Australia). After adding non-reducing loading dye, samples were separated by 15% SDS-PAGE. The experiments have been replicated on three independent occasions.

### 2.8. Cell-Based Assay

For cell-based assays, coding sequences of WT DsbA and its mutant derivatives were subcloned into pBAD322G and introduced into previously constructed UPEC strains CFT073 *ΔdsbA* or CFT073 *ΔdsbA* (pASST) (Appendix A). Cell-based ASST sulfotransferase enzyme assays were performed as previously described [32]. Briefly, overnight cultures of CFT073 and CFT073 *ΔdsbA* carrying DsbA plasmids grown in LB containing 10 mM of L-arabinose were adjusted to an OD_600_ of 0.4 in fresh LB media. Bacterial cell cultures (60 µL) were then mixed with 10 mM of phenol and 1 mM of potassium 4-methylumbelliferyl sulphate (MUS) in a total volume of 200 µL of LB in a 96-well plate. Plates were immediately monitored spectrofluorometrically (360/450 nm) for 90 min at room temperature in a CLARIOstar plate reader (BMG Labtech, Mornington, Australia).

### 2.9. Protein Crystallisation and Structure Determination

EcDsbA K58A and E24A/K58A were crystallised in the wildtype condition as previously described [32]. Briefly, 1 µL of 30 mg/mL protein was mixed with an equal volume of crystallisation buffer (11–13% PEG8000, 5–7.5% glycerol, 1 mM CuCl_2_, 100 mM sodium cacodylate, pH 6.1) and equilibrated against 500 µL of reservoir buffer at 20 °C using hanging drop vapour diffusion. The crystallisation condition for the other three mutants, E37A, E24A and E24A/E37A/K58A, was identified by high-throughput crystallisation trials. Four commercial screens, including Crystal, Index (Hampton Research, San Diego, USA), JCSG+ and PACT (QIAGEN, Hilden, Germany), were screened in-house using a Crystal Gryphon Crystallisation Robot (Art Robbins Instrument, Sunnyvale, CA, USA). Crystals of E37A, E24A and E24A/E37A/K58A mutants were obtained by mixing 1 µL of 30 mg/mL protein with an equal volume of crystallisation buffer (15–20% PEG8000, 0.1 M phosphate-citrate, pH 3.8–4.4, 0.2 M NaCl) and equilibrated against 500 µL of reservoir buffer at 20 °C using hanging drop vapour diffusion. All crystals were briefly soaked in mother liquor containing 20% glycerol as a cryoprotectant and flash-cooled in liquid nitrogen.

Datasets were collected at the Australian synchrotron on the MX2 beamline equipped with an EIGER X 16M detector. All datasets were indexed and integrated with XDS [33] and scaled using AIMLESS [34]. Phasing was performed by molecular replacement using Phaser [35] and the previously determined structure of EcDsbA as a search model (PDB ID: 1FVK). The final structure was obtained after iterative rounds of manual model building using Coot [36] and refinement in phenix.refine [37]. Data collection and refinement statistics are summarised in Appendix A.

## 3. Results

### 3.1. E. coli DsbA Has a Buried Water Channel Mapping to the Conserved Acidic Region

Considering the observed conservation of an acidic groove in the non-catalytic side of DsbA proteins, we carried out a detailed inspection of this area in EcDsbA (PDB: 1FVK). This revealed that three residues, E24, E37 and the positively charged K58 located close to the catalytic site (<13 Å) (Figure 1D), coordinate a network of water molecules (Figure 1D). Our analysis also revealed that this buried water channel connects the thiol group of the buried C33 in the catalytic C30-X-X-C33 motif with the bulk solvent (Figure 1D).

To determine whether these water molecules are commonly present in EcDsbA, we capitalised on the large collection of apo EcDsbA crystal structures available in the PDB database. Of the 36 apo EcDsbA structures solved at a resolution greater than 2.0 Å, 31 structures displayed electron density for water molecules in the buried channel and were selected for this analysis (full list of PDB IDs shown in Appendix A). Four clusters of consensus water molecules, w1, w2, w3 and w4, were identified by overlaying those 31 crystal structures, as illustrated in Figure 1E. Water molecule w1 was observed in 29 out of 31 structures (94%) and formed hydrogen bonds with the backbone amide of the buried C33, and with w2 and w4. The water denoted as w2 was observed in 26 structures (84%) and formed hydrogen bonds with w1, K58 and E37, and was also connected to the bulk solvent at one entrance of the channel. Twenty-eight structures (90%) contained w3, which formed a hydrogen bond with E37 and connected to the bulk solvent at another entrance of the channel. Finally, w4, the least conserved water molecule found in 13 structures (42%), formed hydrogen bonds with w1 and E24. In addition to the water-mediated hydrogen bond network, in some structures E37 and K58 were within salt bridge interaction distances, which would further stabilise the charged residues and the hydrogen bond network. Conserved bulk water (bw1 and bw2) was also found at the entrance of the channel, which coordinated with the solvent-exposed E37. Indeed, 34 out of the 36 apo EcDsbA crystal structures (94%) have one or both bulk water molecules located at a hydrogen bond distance (<3 Å) from E37.

Although DsbA proteins share a high structural similarity, their amino acid sequence conservation is low [15,38]. They have been classified into four phylogenetic subgroups, Ia, Ib, IIa and IIb [39]. To assess the prevalence of the charged residues neighbouring the catalytic site, we carried out a sequence alignment of members of DsbA that have been structurally and biochemically characterised [39]. Specifically, we selected 15 DsbA proteins and used T-coffee Expresso, which allowed a structure-based sequence alignment [40] (Figure 2). Sequence comparison showed that E24 (*E. coli* DsbA numbering) is strictly conserved in class Ia and Ib DsbAs, with more variation observed in DsbA-II. Similarly, E37 is also highly conserved in class Ia and Ib DsbAs, with some of the latter having an aspartic acid residue. The limited set of Class II DsbA sequences shows slightly more variability in this position, which can be a glutamic acid, other charged residues such as aspartic acid and histidine or polar residues such as asparagine. K58 is also highly conserved in class I DsbA, with some having the lysine substituted with a positively charged arginine or threonine. Overall, the negatively charged groove present on the non-catalytic face of DsbA seems to be a conserved feature of most DsbAs (Figure 2B–E). We also analysed in detail the divergent DsbA structures and uncovered that all DsbA structures possessed a buried pocket behind the active site, with the w1 water molecule being strictly conserved (Figure 2F–I). The proximity of the charged residues and coordinated water molecules to the active site and their conservation across diverse DsbA proteins led us to hypothesise that they may have a functional role.

### 3.2. EcDsbA Acidic Region Mutants Display Similar Redox Properties to Wildtype EcDsbA

To probe the function of these conserved charged residues, we generated a set of DsbA mutants where the side chains of E24, E37 and K58 that directly participate in water coordination in the crystal structure of oxidised EcDsbA (PDB ID: IFVK) [41] were substituted with alanine, to remove their charge and water-coordinating capability. Five DsbA mutants, E24A, E37A, K58A, E24A/K58A and E24A/E37A/K58A, were constructed using site-directed mutagenesis. DsbA wildtype and mutants were expressed and purified as previously described [42]. CD spectroscopy wavelength scan results showed that the overall fold and distribution of secondary structural elements were not affected by the introduced mutations (Appendix A).

We next assessed whether the mutations introduced in the charged residues neighbouring the EcDsbA redox centre affected the redox properties of EcDsbA. With a reduction potential of −122 mV, EcDsbA is one of the strongest thiol oxidants reported to date [43]. We determined the redox potential of all generated mutants by measuring their equilibrium constants (K_eq_) with glutathione using a value of −240 mV for the standard potential of the GSH/GSSH redox pair [44]. Under our experimental conditions, the positive control EcDsbA yielded a E_0_^’^ of −125 ± 1 mV, similar to the previously reported value (−122 mV) [43]. All single mutants, E24A, E37A and K58A, and the double mutant E24A/E37A, displayed almost identical redox potential to the wildtype (Figure 3A, Table 1). Conversely, introduction of the three mutations, E24A/E37A/K58A, resulted in a less oxidising redox potential of −139 ± 1 mV, 14 mV lower than that of wildtype.

The strong oxidation activity of DsbA primarily stems from the extremely low p*K*_a_ of the nucleophilic C30 thiol (~3.4) as well as the increased stability of the reduced over the oxidised form [45]. We therefore investigated the p*K*_a_ of the C30 in the mutants by measuring a thiolate-specific increase in absorbance at 240 nm at various pH values using a high-throughput approach [18]. The positive control DsbA wildtype yielded a p*K*_a_ of 3.3 ± 0.1, which is similar to the value reported in the literature [46]. The p*K*_a_ values of the five mutants fell in a narrow range of 3.3–3.7 ± 0.1 (Figure 3B). This result indicated that alanine mutations of the buried charged residues did not affect the reactivity of the nucleophilic cysteine.

We also examined the impact of alanine mutations on the thermal stability of oxidised and reduced DsbA by CD spectroscopy (Appendix A). The melting temperature (T_m_) of these proteins was determined using the Gibbs–Helmholtz equation and is summarised in Table 1. The results showed that the mutations did not significantly affect the overall thermal stability of the protein, whereby all mutants were more stable in the reduced state relative to the oxidised state.

### 3.3. Mutation of Charged Residues in EcDsbA Alters the Conserved Water Molecule Network

To further investigate the charged residues and the connective water network they form, we evaluated the impact of alanine mutations of the residues E24, E37 and K58 on the protein structure and water connectivity by X-ray crystallography. Crystal structures of all five mutants (E24A, E37A, K58A, E24A/K58A, E24A/E37A/K58A) were solved at resolutions ranging between 1.47 and 2.3 Å. DsbA K58A and double-mutant E24A/K58A were crystalised in space group C2, with two molecules per asymmetric unit. The other three mutants, E37A, E24A and E24A/E37A/K58A, were crystallised in space groups P2_1_, P2_1_2_1_2_1_ and P2, respectively, with four molecules per asymmetric unit. All structures were refined to crystallographic R_free_ values of 0.1838–0.2588 (Appendix A). Despite the different crystallisation conditions and space groups, all mutants adopted similar three-dimensional folds (Figure 4A). Overlay of chain A of wildtype DsbA (PDB: 1FVK) and chain A of E24A/E37A/K58A revealed the largest structural variation among all mutants, with an RMSD of 0.53 Å over 174 Cα atoms. In comparison, overlay of chain A of wildtype DsbA and chain A of the E24A/K58A mutant showed the smallest difference, with an RMSD of 0.34 Å over 174 Cα atoms (Appendix A).

We then analysed the impact of these mutations in the buried water channel. Water w4 was not observed in the crystal structures of any DsbA mutants, which is consistent with the relatively low conservation (42%) of this water molecule in the wildtype structures (Figure 4B–H). The E24A mutant lacked the water-coordinating carboxyl group of the glutamic acid, which abolished the hydrogen bonding with water w3 and caused additional water molecules (w1’, w2’ and w3’) to occupy the space filled by the glutamic acid side chain in the native structure (Figure 4D). In this structure, the K58 side chain shifts towards the E24-vacated space, and this change disrupts the proposed salt bridge between K58 and E37. The latter salt bridge was also abolished in the E37A mutant. The hydrogen bond between E37 and water w3 was removed in the E37A mutant structure which has the effect of shortening the distance between w1 and w3 from 3.7 Å in the wildtype structure to 3.3 Å in the mutant structure (Figure 4E). Furthermore, the E37A mutation abolished the interaction of E37 with bulk water. Overall, in the E24A and E37A structures, although water molecules w1, w2 and w3 were still present and coordinated to the remaining charged residues in the channel, the number of interactions stabilising these water molecules was reduced, with the E37A mutant also resulting in the water network being less tightly connected to the bulk water (bw1/bw2).

The crystal structures of the three mutants containing the K58A mutation showed that the shorter alanine side chain at position 58 significantly expanded the size of the entrance to the buried channel. In all three crystal structures, a glycerol molecule, used for cryoprotection, was modelled into the electron density (Figure 4F–H). The binding of glycerol displaced w2 in all three structures and hindered precise analysis of the water network in these mutants. However, examination of the three lysine mutant structures showed that w1 was strictly conserved, forming hydrogen bonds to the backbone amide of C33. When E37 was present in the protein structures, water w3 was observed to form a hydrogen bond with E37 in both K58A and E24A/K58A structures. Additional water molecules (labelled as w’ in Figure 4F–H) were modelled into the electron density in the buried channel in all the lysine mutants. Overall, introduction of mutations into the EcDsbA-charged groove, particularly where more than one amino acid was replaced, appeared to increase the number of solvent molecules bound in the groove, with concomitant loss of the tightly structured water network that connects the catalytic C33 with the bulk solvent in the native structure.

### 3.4. Acidic Region Mutations Compromises DsbA-Catalysed Oxidation of Substrates

We next sought to determine the functional effect of the alanine substitutions in the charged buried groove of EcDsbA using a fluorescence resonance energy transfer (FRET)-based substrate oxidation assay. In this assay, oxidised DsbA protein catalyses the disulphide bond formation between the two terminal cysteines of an unfolded peptide whose sequence is derived from the substrate protein PilQ [47]. Cyclisation of the peptide by disulphide bond formation brings the peptide N-terminal DOTA-europium and the C-terminal fluorescent coumarin into close proximity, resulting in an increase in the fluorescence signal. As shown in Figure 5A, all mutations compromised DsbA-mediated oxidation of the PilQ-derived peptide. E37A caused the most dramatic decrease in the initial rate among single mutants, which was 6-fold slower than that of the wildtype (Table 2). E24A caused a 2.9-fold reduction in the initial rate, while K58A resulted in a 1.9-fold reduction in the initial rate. The double (E24A/K58A) and the triple (E24A/E37A/K58A) mutants showed similar activity to that of E37A. Similar results were obtained when we used a different peptide sequence derived from the protein ASST [48] (Table 2 and Appendix A).

To obtain more detailed kinetic characterisation of the oxidation activity of the different mutants, we next used a stopped-flow fluorescence assay (Figure 5B). The oxidation rate constant catalysed by DsbA wildtype for a peptide derived from substrate protein PapD [49] was measured under the second-order condition and was determined to be 2.26 ± 0.04 × 10^6^ M^−1^ s^−1^ (Table 2). This result is comparable to the rate constant previously reported for DsbA-mediated oxidation of substrate ASST (4.94 × 10^5^ M^−1^ s ^−1^) [50] and falls within the range of rate constants reported for DsbA-catalysed oxidation (10^5^–10^7^ M^−1^ s^−1^) [51]. In agreement with PilQ and ASST peptide oxidation assays, E37A caused the most significant effect among the single mutants, whereby it catalysed PapD peptide oxidation with a rate constant of 1.68 ± 0.12 × 10^6^ M^−1^ s ^−1^, which is ~26% slower than the wildtype EcDsbA (Figure 5B). Intriguingly, E24A yielded a rate constant of 2.24 ± 0.14 × 10^6^ M^−1^ s ^−1^, almost identical to the wildtype. The K58A mutant exhibited an oxidation rate of 2.10 ± 0.17 × 10^6^ M^−1^ s ^−1^, which is also not statistically different from the wildtype. However, the double-mutant E24A/K58A yielded a rate constant of 1.47 ± 0.06 × 10^6^ M^−1^ s ^−1^, 35% slower than that of wildtype EcDsbA. These results indicated that E24A and K58A did influence DsbA oxidase activity, but its individual effect may be too small to be detected in this assay. Modification of all three charged residues (E24A/E37A/K58A) led to an oxidation rate constant of 5.93 ± 0.16 × 10^5^ M^−1^ s^−1^, 3.8-fold (380%) slower than that of EcDsbA wildtype and the slowest among all mutants (Figure 5B).

Collectively, these assays showed that all three charged residues, E24, K58 and particularly E37, contributed to DsbA oxidation activity, and that removal of all three charged residues resulted in a reduction of the catalytic activity of EcDsbA. In line with *in vitro* data, E37A, E24A/E37A and E24A/E37A/K58A also affected *in vivo* enzymatic activity of plasmid-expressed periplasmic ASST, which was previously described to require DsbA for functional folding [50] ( Appendix A). Intriguingly, the effect of individual mutation E24A was undetectable in this *in vivo* assay.

### 3.5. Mutations Did Not Slow the Resolution of Mixed-Disulphide Intermediate

Earlier studies showed that mutations introduced in residues adjacent to the active site can slow the resolution of the covalent mixed-disulphide intermediate formed between DsbA and substrates [52]. To investigate whether the mutants generated in this work affected the release of the substrate, we mixed DsbA mutants and PapD-derived peptide at a 1:1.1 molar ratio and quenched the reaction with trichloroacetic acid (TCA) at different time points (10 s, 20 s, 30 s and 120 s). To differentiate oxidised DsbA, reduced DsbA and the DsbA-peptide intermediate in the SDS-PAGE analysis, the sample was treated with thiol alkylating agent 4-acetamido−4’-maleimidylstilbene−2,2’-disulfonic acid (AMS, ~500 Da). Oxidised DsbA does not have free thiols, so it cannot be alkylated and remains unmodified. Alkylation by AMS of reduced DsbA (two free thiols) resulted in a ~1 kDa shift in mass relative to oxidised DsbA. Alkylation of the DsbA-peptide intermediate resulted in a ~1.5 kDa shift relative to oxidised DsbA. As shown in Figure 5C, no bands corresponding to the intermediate complex were observed for the wildtype, the double E24A/K58A or the triple E24A/E37A/K58A mutant samples collected at different time points. This demonstrated that the mutations generated in this work did not seem to slow the resolution of the mixed-disulphide intermediate. However, at the 10 s and 20 s time points, the double and triple mutants contained more oxidised DsbA relative to the wildtype, indicating that the DsbA-mediated oxidation process was less efficient, in line with the observation from the kinetic studies.

## 4. Discussion

In the periplasm of Gram-negative bacteria, the disulphide bond (DSB) enzyme system mediates disulphide formation, with DsbA being the key bacterial oxidative folding catalyst [3]. Although related to the archetypal disulphide reductase enzyme thioredoxin [53], bacterial DsbAs display differences in sequence and structural topology, which result in these proteins being electron acceptors rather than donors [3]. DsbA proteins have been extensively investigated, particularly the *E. coli* prototype EcDsbA, a reference thioredoxin-like enzyme that epitomises the structural and redox features present in efficient thiol oxidases. Despite our detailed knowledge on EcDsbA, certain structural features have remained less well-understood, for example the significance of a conserved charged region mapping to the non-catalytic face of DsbA proteins. In this work, we aimed to dissect whether the charged residues in this region (E24, E37 and K58 in EcDsbA), which coordinate a buried water network connecting the catalytic site with the bulk solvent, served a functional role.

Previous comprehensive studies had attributed EcDsbA efficient thiol oxidase activity to a highly oxidising redox potential, the low p*K*_a_ of C30 and the increased stability of the reduced form of the protein [45,54,55,56]. To investigate the influence of the charged residues on the redox properties and reactivity of EcDsbA, we generated a series of mutants containing E24A, E37A and K58A mutations. The structures of these mutants exhibited similar structural characteristics to the wildtype EcDsbA structure, as shown by CD spectroscopy and X-ray crystallography. Furthermore, they all had comparable redox properties to the unmodified enzyme, including similar nucleophilicity of the catalytic cysteine (similar C30 thiol p*K*_a_ values), a more thermally stable reduced form and redox potential values similar to the native protein (only the triple-mutant E24A/E37A/K58A yielded a lower redox potential, Table 1).

Despite the structural conservation and the comparable redox properties, functional characterisation of DsbA mutants showed that the alanine mutations of E24, E37 and K58 had pronounced effects on DsbA catalytic activity. This effect was particularly evident for the E37A mutant, which showed compromised *in vitro* activities in two different assays with distinct substrate peptides. A significant effect was also observed for the E24A/K58A double mutant and the E24A/E37A/K58A triple mutant, which also exhibited reduced oxidising activity towards divergent peptides derived from PilQ, ASST and PapD substrates. Cell-based assays showed that all DsbA mutants except E24A significantly affected ASST sulfotransferase activity *in vivo*, consistent with our *in vitro* data.

A notable observation of this work was that although the introduced alanine mutations did not significantly change the redox properties of DsbA, some mutations considerably altered DsbA oxidase activity (only the triple mutant displayed a more reducing redox potential). To understand the molecular basis for this apparent discrepancy, we considered the DsbA-dependent disulphide formation process. Mechanistically, this reaction is thought to occur in two steps [57,58,59], whereby first a substrate cysteine in the thiolate form attacks the sulphur atom of the N-terminal cysteine of the oxidised CXXC active site (C30 in EcDsbA), resulting in the formation of an intermediate disulphide-bonded DsbA-substrate complex (Figure 6). For the reaction to proceed in the forward direction and avoid futile redox cycles, the DsbA C-terminal cysteine (C33 in EcDsbA) must be stabilised [55], probably by protonation, allowing another deprotonated cysteine from the substrate to resolve the intermediate complex, which then results in the release of the oxidised substrate and reduced DsbA (Figure 6).

Wildtype DsbA structures had revealed that E24, E37 and K58 directly coordinate a network of water molecules (w1–w4) in a buried channel extending from the bulk solvent to the active site. Extensive hydrogen bond networks between water molecules and the polar side chains of residues have been previously identified in the cavities of a large number of proteins, such as cytochrome c oxidases [60], bacteriorhodopsin [61] and isatin hydrolase [62]. Furthermore, it has been suggested that these water-coordinating polar/charged residues can either directly relay protons from bulk water or modulate the proton transfer in water chains inside protein grooves [63]. The hydrogen-bonding network between water molecules and residues E24, E37 and K58 in the buried channel of DsbA may serve as a Grotthuss-type proton wire [62,63] to relay protons to the active site centre and facilitate the disulphide bond catalysis. Previous work has shown that pathways mediated by different residue types influence proton relay efficiency [62]. We therefore reasoned that the alanine mutants that we generated affected the catalytic activity of DsbA, not by affecting its redox properties but possibly by compromising its water-coordinating capability.

To investigate whether the water network had been altered upon the introduction of the alanine mutations, we inspected the DsbA alanine mutant crystal structures, which showed that although all point mutants maintained some of the w1–w4 molecules, their position was altered and the tight connectivity of waters to stabilising charged amino acids was considerably reduced. This was especially the case for E37A, which lacked the negatively charged side chain stabilising the buried water network and connecting it with the bulk solvent, which could contribute to initiation of the proton relay. We also found that this mutant showed the lowest oxidase activity of all the single mutants we analysed. Mutation of two or more of the E24, E37 and K58 further increased the “disordered” solvent content, reducing the structured hydrogen bond network between the charged residues and the w1–w4 water molecules. These changes would cause a significant disruption to the proposed proton wire and affect the proton relay to the active site C33, and consequently impact on the oxidase activity. Indeed, both E24A/K58A and E24A/E37A/K58A mutants displayed significantly lower thiol oxidase activity compared to the wildtype protein (although the more reducing redox potential of the latter would also affect its oxidase activity).

Inefficient protonation and stabilisation of the C33 thiolate could lead to the DsbA-substrate complex intermediate being resolved by both DsbA C33 or the substrate-free thiolate (Figure 6). The first scenario would reverse the oxidation reaction to produce the initial reagent-oxidised DsbA and the reduced substrate. Our electron transfer assays (Figure 5C) provided indirect evidence for this occurrence. While native DsbA rapidly oxidised the substrate peptide (in less than 10 s), in the case of the DsbA double E24A/K58A and triple E24A/E37A/K58A mutants, oxidised DsbA was present in the reaction for over 10 and 20 s, respectively. The observed accumulation of oxidised DsbA could be a result of futile oxidation cycles that restore oxidised DsbA or could derive from a compromised substrate-binding affinity of the mutant proteins compared to the native. The comparable structures and redox properties of the DsbA mutants with wildtype DsbA however make the latter hypothesis less likely.

Collectively, our results indicate that charged residues in the non-catalytic face of DsbA are important for its activity. We propose that the charged EcDsbA E24/E37/K58 residues form a buried water network which serves as a proton wire to relay protons from the bulk solvent to the active site C33 and facilitate the forward direction of the disulphide bond catalysis. We cannot completely rule out the possibility that alanine mutations of the charged residues may impede other functions of DsbA or affect the binding affinity of substrate peptides, but considering the unperturbed structures of the mutants, the similar redox properties and the in silico evidence of proton shuffling in a DsbA homologous protein [64], disruption of the proton wire seems to be the most plausible explanation for the reduction of enzyme activity. It is commonly accepted that the titratable amino acids are required in proton pathways to facilitate proton uptake [65]. In systems such as human voltage-gated proton channel Hv1 and cytochrome c oxidase, charged residues serve as a selectivity filter that only allows protons to enter the pathway [66,67]. In the case of EcDsbA, only polarity seems to be required for proton transfer, since a previous study showed that E24Q, E37Q and K58M mutations, which removed the charge but maintained the size and polarity of the side chains, did not affect the reactivity of EcDsbA *in vitro* or *in vivo* [68]. Nonetheless, sequence alignment of diverse DsbA homologues showed that charged residues in these positions are favoured among DsbA proteins.

The present work could have broader implications for other TRX-like proteins, as the buried water channel identified across DsbA homologues is also present in other TRX-like proteins. Indeed, proton relay mechanisms have been proposed for other thioredoxin family proteins. For example, a molecular dynamics simulation of DsbL postulated a proton exchange between the buried active site cysteine and neighbouring K23 (equivalent to E24 in EcDsbA), whereby K23 directly donates a proton to stabilise the thiolate form of the buried C32 upon substrate oxidation and accepts a proton from C32 when DsbL is re-oxidised by the partner oxidase DsbI [64]. In the case of the reductase thioredoxin, the archetype of the TRX family, D26 and K57 (equivalent to E24 and K58 in EcDsbA) were shown to influence the redox activity [23] and the rate of resolution of TRX-substrate intermediates [69] (Appendix A). Similarly, E200 of the eukaryotic PDI protein ERp72 (equivalent to E24 in EcDsbA) has also been suggested to transfer protons to the C-terminal active site cysteine to modulate PDI activity [70,71].

## 5. Conclusions

Overall, our detailed analysis of the EcDsbA catalytic mechanism, together with previous studies, suggested a common thiol-disulphide exchange reaction mechanism in the large TRX protein superfamily, whereby a proton relay system mediates proton donation (in oxidases) or subtraction (in reductases) to the buried C-terminal active site cysteine to drive the reaction in a forward direction and prevent futile backwards redox cycles. In the case of DsbA proteins, considering their pivotal role in bacterial virulence [72,73], the identified functional groove sandwiched between the thioredoxin and helical domains represents a new potential allosteric inhibitory site for these enzymes. The DsbA mutants in this work retained a key water molecule next to the catalytic cysteine as well as some redox activity, which may indicate that the designed allosteric inhibitors would require to completely displace/block all water molecules in the groove for a significant reduction of DsbA function. In this context, a study on *Acinetobacter baumannii* DsbA [14] may have serendipitously demonstrated the suitability of this inhibitory site as it showed that the prokaryotic elongation factor EF-Tu, which binds strongly to the charged residues on the non-catalytic site of AbDsbA (Appendix A), inhibited its oxidase activity *in vitro*. The mechanistic understanding gained in our study could therefore inform the design of new classes of inhibitors against DsbA enzymes.

## Figures and Tables

**Figure 1 antioxidants-12-00380-f001:**
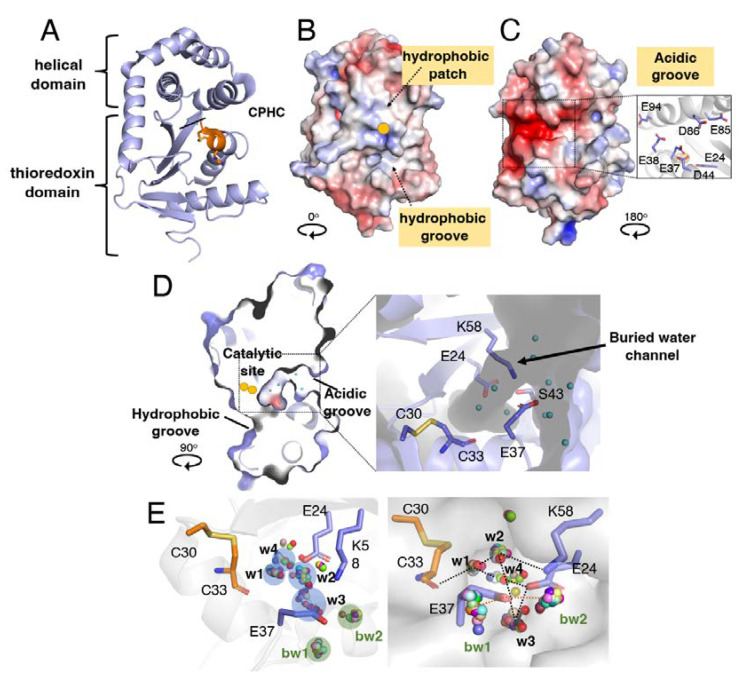
Structure and water network in a buried channel of EcDsbA. (**A**) Crystal structure of oxidised *E. coli* DsbA represented as a ribbon model (PDB: 1FVK). The catalytic site “C^30^PHC^33^” is coloured in orange. Electrostatic surface presentation of catalytic face (**B**) and non-catalytic face (**C**) of oxidised EcDsbA. The solvent-exposed active site C30 is shown as a yellow sphere. Inset in (**B**) shows the acidic residues present in the acidic groove of EcDsbA. (**D**) Cross-section of EcDsbA structure rotated clockwise by 90° relative to the catalytic face shown in A. The positions of active site residues C30 and C33 are highlighted by yellow spheres. Inset shows a close-up view of a narrow water channel buried in the core of EcDsbA protein surrounded by the charged residues E24, E37 and K58. (**E**) Four conserved water molecules (w1, w2, w3 and w4) in the buried cavity and two conserved bulk water molecules (bw1, bw2) coordinated to E37 were identified by clustering of water molecules in 30 superimposed apo EcDsbA structures with resolution better than 2 Å. Clusters of water molecules are shown in two different orientations in the left and right panels. Blue/green-shaded circles indicate the different clusters of water molecules. Water molecules distant from the circles are regarded as outliers. EcDsbA is presented in a ribbon model in the left panel, and the surface of EcDsbA is shown in the right panel. Hydrogen bonds are indicated by black (w1–4) or orange (bw1–2) dashed lines in the right panel.

**Figure 2 antioxidants-12-00380-f002:**
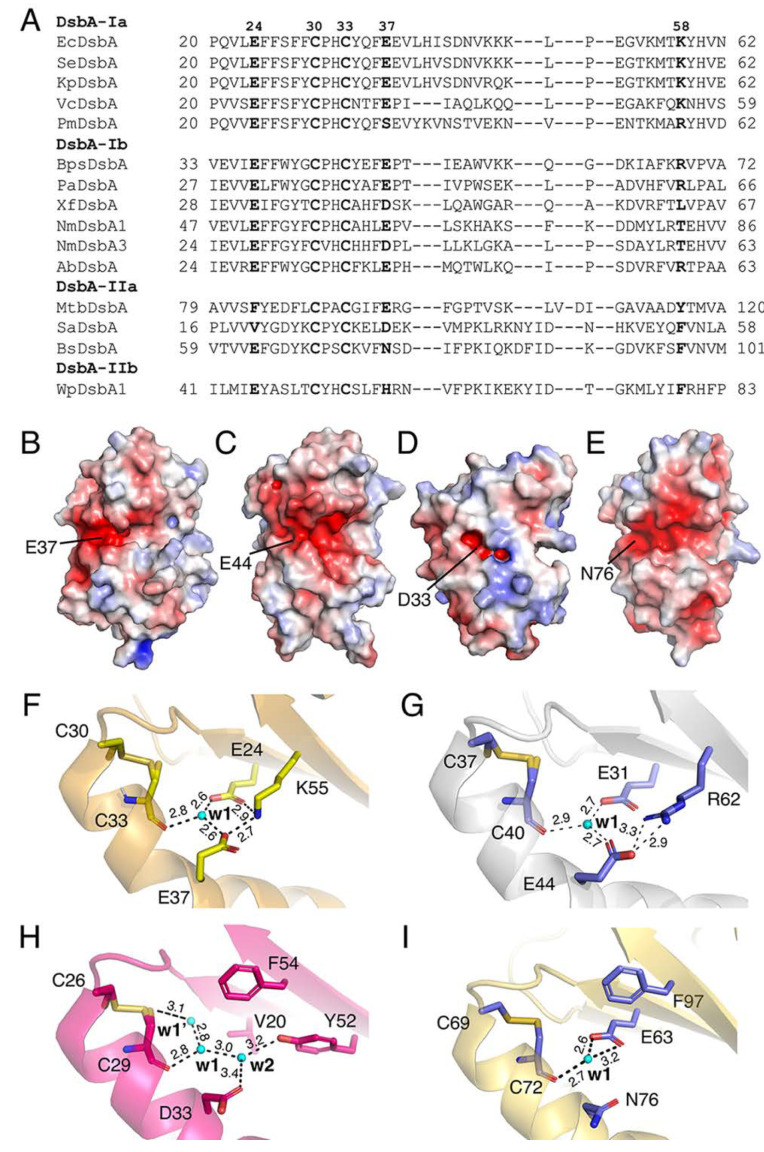
Conservation of charged residues, charged patch and water-mediated proton wire in DsbA proteins. (**A**) Alignment of representative DsbA from each structural class. (**B**–**E**) Electrostatic surface representation of the non-catalytic surface of DsbA from *E. coli* (PDB ID: 1FVK) (**B**), *P. aeruginosa* (PDB ID: 3H93) (**C**), *S. aureus* (PDB ID: 3BCI) (**D**) and *B. subtilis* (PDB ID: 3EU3) (**E**). Electrostatic surface potential is contoured between −5 (red) and +5 (blue) kT/e. Solvent-exposed E37 of EcDsbA and its equivalents in other DsbAs are labelled. (**F**–**I**) Possible water-mediated proton wire present in *V. cholera* DsbA (**F**), *P. aeruginosa* DsbA (**G**), *S. aureus* DsbA (**H**) and *B. subtilis* DsbA (**I**). In each case, a bound water labelled w1 forms hydrogen bonds with the buried C-terminal active site cysteine and charged residues similar to those observed in EcDsbA. Unlike the structure of EcDsbA which has 3–4 water molecules in the buried channel, most DsbA structures from other bacteria only had one water molecule, w1. In *V. cholera* DsbA, w1 formed hydrogen bonds with C33, E24 and E37. K55 (equivalent to K58 in EcDsbA) did not directly coordinate to w1 but stabilised the network by hydrogen bonding to E24 and E37 (**F**). Similarly, w1 in *P. aeruginosa* DsbA coordinated to the buried active sites C40, E31 (equivalent of E24 in EcDsbA) and E44 (equivalent of E37 in EcDsbA). R62 (equivalent to K58 in EcDsbA) formed hydrogen bonds with E44 (**G**). *S. aureus* DsbA had two additional water molecules, w1’ directly coordinated to the free thiol of the C29 (C33 equivalent) in the reduced form and w2 coordinated to D33 (equivalent to E37 in EcDsbA) and Y52. The equivalent position of E24 in EcDsbA was V20 in SaDsbA, which cannot form hydrogen bonds. Its role seems to be replaced by Y52, which directly coordinates to w2 (**H**). In *B. subtilis* DsbA, the E37 residue of EcDsbA is replaced by the polar residue N76. Despite its water-coordinating capability, it did not form hydrogen bonds with w1 in the crystal structure. Instead, w1 coordinated to the buried active sites C72 and E63 (equivalent to E24 in EcDsbA) (**I**).

**Figure 3 antioxidants-12-00380-f003:**
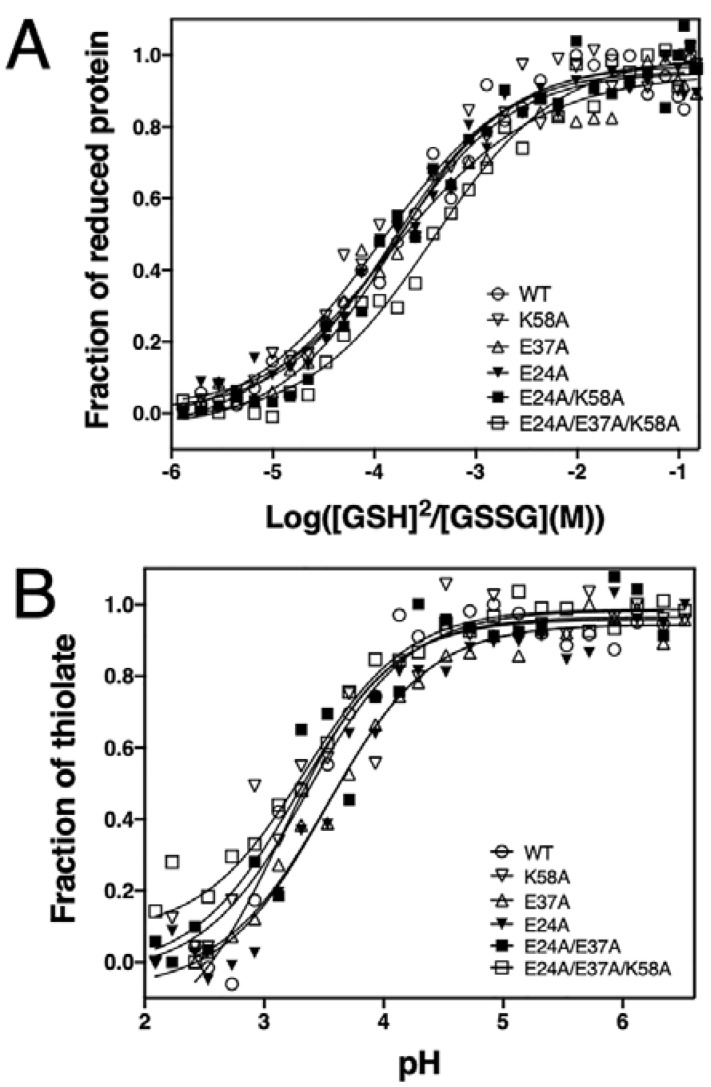
Redox characterisation of EcDsbA mutants. (**A**) Redox potential of EcDsbA mutants. Experiments were repeated on three independent occasions and a representative curve is presented for each DsbA mutant. The redox potential values of all DsbA mutants are listed in Table 1. (**B**) p*K*_a_ values of the nucleophilic C30 of EcDsbA mutants. Experiments have been repeated on three independ-ent occasions and only a representative curve for each DsbA mutant is shown. The p*K*_a_ values of the C30 of all DsbA mutants are listed in Table 1.

**Figure 4 antioxidants-12-00380-f004:**
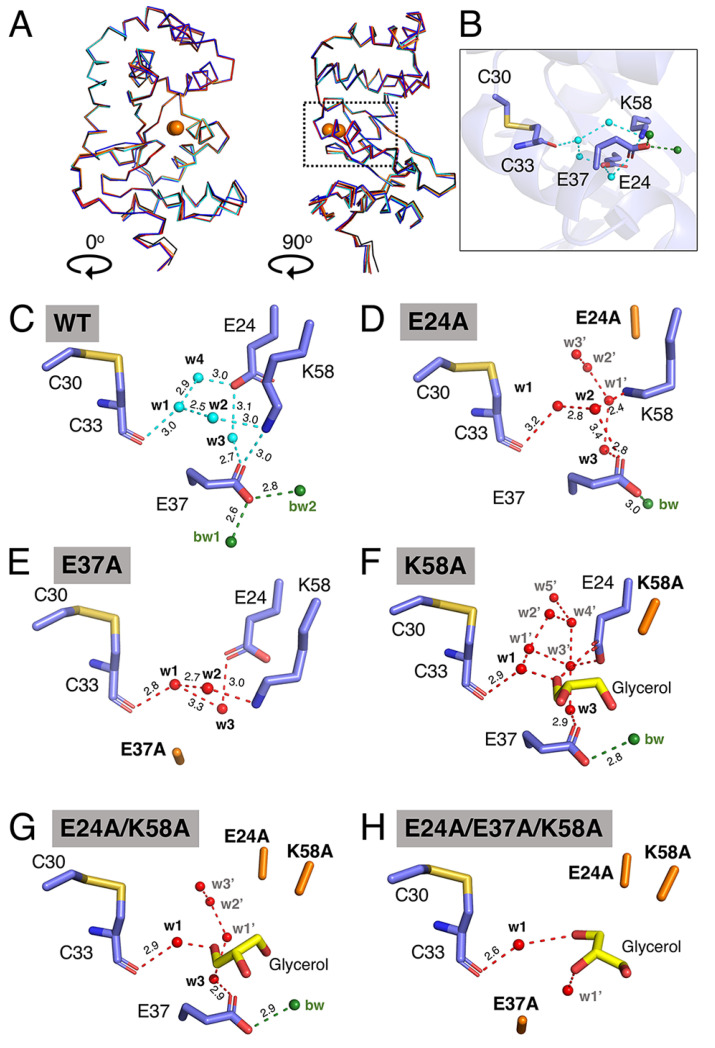
Structural characterisation of EcDsbA mutants by X-ray crystallography. (**A**) Structural overlay of EcDsbA wildtype (blue, PDB ID 5QO9), E24A (salmon), E37A (orange), K58A (cyan), E24A/K58A (red) and E24A/E37A/K58A (black). The overlaid structures are presented in two orthogonal orientations, labelled 0° and 90°. Active site cysteines are shown as orange spheres. A dashed box indicates the location of the expanded view of EcDsbA wildtype shown in (**B**). (**C**–**H**) Water network in buried water channel of EcDsbA wildtype, mutants E24A, E37A, K58A, E24A/K58A and E24A/E37A/K58A. Hydrogen bonds are shown as dashed lines and distances in Å are indicated. Water molecules are shown as cyan or red spheres and labelled. Bulk water (bw) is shown as green spheres.

**Figure 5 antioxidants-12-00380-f005:**
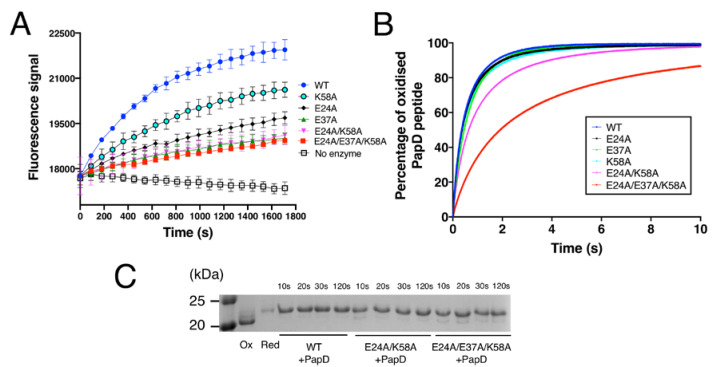
Functional characterisation of EcDsbA mutants. (**A**) Thiol-disulphide oxidase activity of EcDsbA mutants was determined by monitoring the oxidation of a fluorescently labelled peptide derived from substrate PilQ. Data points shown in the figure are mean ± standard error of the mean (SEM), *n* = 3. (**B**) Kinetics of the disulphide exchange reaction between EcDsbA mutants and a 10-mer peptide derived from substrate PapD at pH 7.0 and 25 °C. All reactions were performed in a stopped-flow fluorescence instrument under the second-order condition with initial concentrations of 1 µM for EcDsbA and 1 µM for PapD-derived peptide. Kinetics of reactions were determined by monitoring the increase in fluorescence upon reduction of EcDsbA. Experiments were repeated on three independent occasions and a representative curve is presented for each DsbA mutant. (**C**) Electron transfer assay for EcDsbA mutants and the PapD-derived peptide. Oxidised EcDsbA mutants and reduced PapD-derived peptide were mixed in a 1:1 molar ratio. Samples were taken from the reaction mixture after 10 s, 20 s, 30 s and 120 s and treated with 10% TCA. Free cysteines were labelled with AMS and samples were analysed by SDS-PAGE. Only part of the gel is shown for clarity, the full SDS-PAGE gel is shown in Appendix A.

**Figure 6 antioxidants-12-00380-f006:**
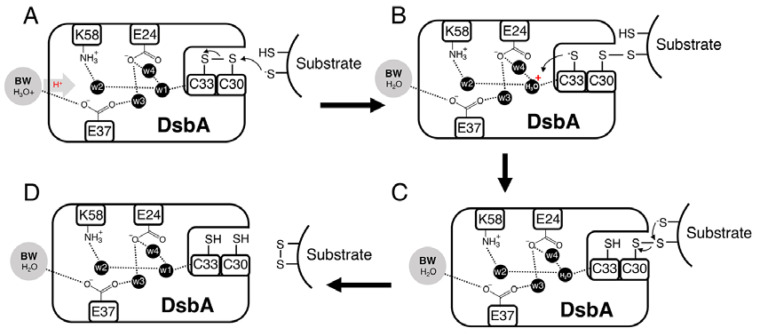
Proposed catalytic mechanism of DsbA. (**A**) Protonated bulk water (BW) donates a proton through the hydrogen-bonded proton wire to w1, which is hydrogen-bonded to the backbone amide of the DsbA *C*-terminal cysteine (C33 in EcDsbA). (**B**) A substrate cysteine in the thiolate form attacks the sulfur atom of the *N*-terminal cysteine of the oxidised CXXC active site (C30 in EcDsbA), resulting in the formation of an intermediate disulphide-bonded DsbA-substrate complex. For the reaction to proceed in the forward direction and avoid futile redox cycles, stabilisation/protonation of the DsbA *C*-terminal cysteine (C33 in EcDsbA) is required. We propose this could be achieved by C33 abstracting a proton from the nearby w1. (**C**,**D**) The mixed disulphide is then resolved by a deprotonated substrate cysteine, after which the oxidised substrate and reduced DsbA are released.

**Table 1 antioxidants-12-00380-t001:** Redox and biochemical parameters of DsbA mutants. p*K*_a_ and *K*_eq_ are expressed as mean ± standard error of the mean (SEM) with at least three independent replicates.

EcDsbA Variants	p*K*_a_ of the Cys30 Thiol	*K*_eq_ (M)	E’_0_ (mV)	T_m_ (°C)		References
				Oxidised	Reduced	
Wildtype (literature)	3.55 ± 0.02	1.08 ± 0.02 × 10^−4^	−123	69.3	77.5	(34, 37, 48)
Wildtype	3.3 ± 0.1	1.65 ± 0.11 × 10^−4^	−125 ± 0.4	67.4	75.7	This study
E24A	3.7 ± 0.1	1.37 ± 0.29 × 10^−4^	−124 ± 3	63.9	74.6	This study
E37A	3.3 ± 0.1	1.36 ± 0.07 × 10^−4^	−124 ± 0.5	65.0	74.1	This study
K58A	3.4 ± 0.1	1.14 ± 0.01 × 10^−4^	−121 ± 0.03	63.2	70.4	This study
E24A/K58A	3.5 ± 0.1	1.64 ± 0.10 × 10^−4^	−126 ± 0.8	60.6	70.4	This study
E24A/E37A/K58A	3.5 ± 0.1	3.88 ± 0.48 × 10^−4^	−139 ± 1	66.7	74.3	This study

**Table 2 antioxidants-12-00380-t002:** Kinetic parameters of DsbA mutants. Initial rate and rate constants are expressed as mean ± standard error of the mean (SEM) with at least three independent replicates.

EcDsbA Variant	Substrate		
	PilQ-Derived Peptide	ASST-Derived Peptide	PapD-Derived Peptide
	Initial Rate by POA (RFU/s)	Initial Rate by POA (RFU/s)	Rate Constant by Stopped Flow (M^−1^ s^−1^)
**Wildtype**	6.67 ± 0.43	79.5 ± 8.4	2.26 ± 0.04 × 10^6^
**E24A**	2.28 ± 0.66	53.6 ± 9.0	2.24 ± 0.14 × 10^6^
**E37A**	1.11 ± 0.55	22.1 ± 14.3	1.68 ± 0.12 × 10^6^
**K58A**	3.55 ± 1.39	72.4 ± 1.9	2.10 ± 0.17 × 10^6^
**E24A/K58A**	1.23 ± 0.28	40.4 ± 15.5	1.47 ± 0.06 × 10^6^
**E24A/E37A/K58A**	1.31 ± 0.46	26.4 ± 6.2	5.93 ± 0.16 × 10^5^

## Data Availability

All data are included in this article or the Appendix A, with the exception of the structure coordinates and structure factors. Structure factors and coordinates have been deposited in the Protein Data Bank (PDB; http://www.pdb.org) under the accession codes 8EQR (E24A mutant), 8EQQ (E37A mutant), 8EQO (K58A mutant), 8EOC (E24A/K58A mutant) and 8EQP (E24A/E37A/K58A mutant).

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
