# Peer review of "A Buried Water Network Modulates the Activity of the Escherichia coli Disulphide Catalyst DsbA"

_antioxidants, 2023, doi:10.3390/antiox12020380_

Round 1

Reviewer 1 Report

My sincere congratulations to all authors for the high quality of their work. Please see my minor points in the attached document.

Author Response

We thank the reviewer for the time dedicated to review this manuscript. please find attached our response. 

Reviewer 2 Report

Excellent biophysical, kinetic and structural study! I really like the proposed water mediated catalytic mechanism. Only one minor comment.  I am not convinced whether this negatively charged non-catalytic site could be a promising site for antibacterial inhibitors because mutating the residues has less than a factor 10 effect on the second order rate constant (Table 2). I suggest to downscale this idea in the paper.

Author Response

We thank this reviewer for their time to review this manuscript. 

We appreciate reviewer's comments. It is correct that even triple mutations in this study did not result in more than a factor of 10 effect. This could be due to several reasons. We also cannot rule out residues other than the three charged residues studied in this work may contribute to proton shuffling. Indeed, the mutations introduced  were unable to displace  water molecule w1, which is conserved in almost all DsbA structures. It is possible that displacement of  this water  by inhibitor binding, maybe also required for a more dramatic effect. It is interesting though that the EF-Tu peptide, which binds to the non-catalytic face of DsbA, reduced dithiol oxidase activity of AbDsbA by 2/3 (Figure S6), strongly suggesting the charged groove on the non-catalytic face of DsbA is a promising inhibitory site. However to address this reviewer’s fair comment we have changed the conclusions  as follows:

Lines 560-564

“The DsbA mutants in this work retained a key water molecule next to the catalytic cysteine as well as some redox activity, which may indicate that the designed allosteric inhibitors would require to completely displace/block all water molecules in the groove for a significant reduction of DsbA function.”